# QTL analysis of traits related to seed size and shape in sesame (*Sesamum indicum* L.)

Hongxian Mei[1,2◉], Chengqi Cui[1,2◉], Yanyang Liu[2], Zhenwei Du[2], Ke Wu[2], Xiaolin Jiang[2], Yongzhan Zheng◉[2]*, Haiyang Zhang[1,2]*

1 The Shennong Laboratory, Zhengzhou, Henan, China, 2 Henan Sesame Research Center, Henan Academy of Agricultural Sciences, Zhengzhou, Henan, China

◉ These authors contributed equally to this work.
* sesame168@163.com (YZ); zhanghaiyang@zzu.edu.cn (HZ)

**Data Availability Statement:** All relevant data are within the paper and its Supporting Information files.

**Funding:** This work was supported by the Natural Science Foundation of Henan (222301420098) to

## Abstract

Seed size and shape are important traits that determine seed yield in sesame. Understanding the genetic basis of seed size and shape is essential for improving the yield of sesame. In this study, $F_2$ and $BC_1$ populations were developed by crossing the Yuzhi 4 and Bengal small-seed (BS) lines for detecting the quantitative trait loci (QTLs) of traits related to seed size and shape. A total of 52 QTLs, including 13 in $F_2$ and 39 in $BC_1$ populations, for seed length (SL), seed width (SW), and length to width ratio (L/W) were identified, explaining phenotypic variations from 3.68 to 21.64%. Of these QTLs, nine stable major QTLs were identified in the two populations. Notably, three major QTLs *qSL-LG3-2*, *qSW-LG3-2*, and *qSW-LG3-F₂* that accounted for 4.94–16.34% of the phenotypic variations were co-localized in a 2.08 Mb interval on chromosome 1 (chr1) with 279 candidate genes. Three stable major QTLs *qSL-LG6-2*, *qLW-LG6*, and *qLW-LG6-F₂* that explained 8.14–33.74% of the phenotypic variations were co-localized in a 3.27 Mb region on chr9 with 398 candidate genes. In addition, the stable major QTL *qSL-LG5* was co-localized with minor QTLs *qLW-LG5-3* and *qSW-LG5* to a 1.82 Mb region on chr3 with 195 candidate genes. Gene annotation, orthologous gene analysis, and sequence analysis indicated that three genes are likely involved in sesame seed development. These results obtained herein provide valuable in-formation for functional gene cloning and improving the seed yield of sesame.

## Introduction

Sesame (*Sesamum indicum* L., 2n = 26) belongs to the *Sesamum* genus under the *Pedaliaceae* family and is one of the earliest oilseed crops to be domesticated [1]. Sesame seed provides high-quality oil containing a high content of unsaturated fatty acids and natural antioxidants for human consumption, and is also traditionally consumed directly [2, 3]. The global demand for sesame seeds and derived products is increasing significantly owing to the shift towards healthier and nutritional plant-based foods [4]. However, the quantity of sesame produced annually is much lower than that of other oil crops, such as groundnut, rapeseed, and sunflower [5]. Therefore, improving the yield of sesame is one of the most important goals of

CC, the China Agriculture Research System (CARS-14–1-01) to YZ, the Central Government-Guided Local S&T Development Fund Project of Henan (Z20221343038) to HM, the Key Research Project of the Shennong Laboratory (SN01-2022-04) to HM, the Key Research and Development Project of Henan (221111520400) to YL, Projects of Henan Academy of Agricultural Sciences (2023ZC081, 2023TD31, 2023JC13) to CC and YL. The funders had no role in the design of the study or the collection, analysis or interpretation of the data or the writing of the manuscript.

**Competing interests:** The authors have declared that no competing interests exist.

sesame breeding [6]. The yield of seeds is significantly affected by seed size and shape; therefore, these factors may have considerable potential for significantly improving the yield of sesame seeds [7]. Additionally, the traits related to seed size and shape have higher heritability and better stability than the traits related to seed yield across different environments [8]. Therefore, dissecting the genetic basis of seed size and shape will aid in improving the yield of sesame seeds.

Quantitative trait locus (QTL) mapping is an effective approach for dissecting the genetic basis of complex quantitative traits of crops. Genetic map is a prerequisite for QTL mapping and can provide essential information regarding the linkage of genetic markers [9]. Several genetic maps of sesame have been constructed in the past decade [10–12]. However, the genetic dissection of the agronomic traits of sesame has been hindered for a long time owing to lack the markers and genomic information [6]. With the release of several physical maps of sesame [13–18] and the development of next-generation sequencing techniques, many QTLs/genes for agronomic traits were identified in sesame using QTL mapping [6, 19–21] and genome-wide association study (GWAS) [4, 22–24]. For QTL mapping, Wu et al. [19] detected 13 QTLs by multiple interval mapping (MIM) and 17 QTLs by mixed linear composite interval mapping (MCIM) for seven grain yield-related traits by a high-density linkage map with 3,769 single-nucleotide polymorphism (SNP) markers. Zhang et al. [20] identified a determinacy gene (*SiDt*) controlling the determinate growth habit using an ultra-dense map in sesame. Du et al. [21] identified 19 major-effect QTLs for seed-related traits using a high-density genetic map with 2,159 SNP markers. Mei et al. [6] constructed a high-density genetic map with 3,528 specific-locus amplified fragment (SLAF) markers, and identified 46 significant QTLs for seven yield-related traits. For GWAS, Wei et al. [22] identified 549 associated loci for 56 traits in four environments by using GWAS. Cui et al. [23] performed GWAS for the seed coat color of 366 sesame germplasm lines in 12 environments and identified 224 SNPs with 1.01–22.10% of phenotypic variation explained (PVE). Sabag et al. [4] identified 50 signals associated with flowering date and yield-related traits. Dossou et al. [24] detected 17 and 72 SNPs associated with sesamin and sesamolin, respectively, and identified 11 candidate causative genes by GWAS.

Several studies have comprehensively analyzed the traits related to seed size and shape in various crops, and numerous QTLs/genes have been identified in rice [25, 26], maize [8, 27], wheat [28, 29], peanut [30, 31], and rapeseed [32, 33]. However, there is a scarcity of studies on the seed size and shape traits in sesame except for Du et al. [21], and the genetic basis of these traits is poorly understood to date. In order to elucidate the genetic basis of these traits in sesame, we developed a $BC_1$ and an $F_2$ population for characterizing the seed size and shape traits in sesame. A high-density genetic map for the $BC_1$ population was constructed using SLAF and SSR markers, and a genetic map for the $F_2$ population was constructed using SSR markers. Three important co-localized loci were subsequently identified, harboring the stable major QTLs, which may provide useful information for future breeding strategies aimed at improving the seed yield of sesame.

## Materials and methods

### Plant materials and phenotypic evaluation

An exotic germplasm line Bengal Small-seed (BS) and a locally adapted elite cultivar Yuzhi4 were crossed for developing the $BC_1$ [6] and $F_2$ segregation populations. The seed size of the male parent BS was significantly smaller than that of the female parent Yuzhi4. The $F_{2:3}$ families were generated by self-pollinating 150 $F_2$ plants. The $F_2$ plants and its $F_{2:3}$ families were grown in Sanya (SY; N18˚14', E109˚29'), Hainan Province, China, in the winter of 2019 and

2020 (hereafter 2019SY and 2020SY), respectively. A total of 150 $BC_1F_2$ families were developed from 150 $BC_1$ plants, and the $BC_1F_2$ families were planted in Pingyu (PY; N32˚59', E114˚42'), Nanyang (NY; N 32˚54', E112˚24'), and Luohe (LH; N 33˚37', E 113˚58'), in Henan Province, China, in the summer of 2018 (hereafter 2018PY, 2018NY, and 2018LH). The traits of the $F_{2:3}$ and $BC_1F_2$ plants were evaluated instead of the $F_2$ and $BC_1$ individuals, respectively, as described in the study by Mei et al. [6]. Each of these locations was regarded as a separate environment. The field experiments were performed in randomized complete blocks with two replicates. Each accession was planted with 20 single plants per row with a spacing of 17 cm between the plants and 40 cm between the rows. Ten uniform plants were harvested from the middle of each row at maturity. The mature seeds from these ten plants were mixed together for measuring the seed length (SL), seed width (SW), and length to width ratio (L/W) using an SC-G automatic seed analysis and 1000-grain weight instrument (Hangzhou Wanshen Detection Technology Co., Ltd. China). The L/W was calculated by dividing the SL by the SW [30]. The mean values of SL, SW, and L/W were calculated from three independent samples of 1000 seeds for phenotypic characterization. The seed size was represented by SL and SW, and the seed shape was represented by L/W.

## Statistical analyses of phenotypic data

The variance and normal distribution of the data were analyzed using the SPSS Statistics software, version 19.0 (IBM Corp., Armonk, NY, USA). The broad-sense heritability ($h^2$) of the traits related to seed size was calculated using the following formula:

$$h_B^2 = \frac{\sigma_G^2}{\sigma_G^2 + \frac{1}{n}\sigma_{GE}^2 + \frac{1}{nr}\sigma_\varepsilon^2}$$

Where $\sigma_G^2$, $\sigma_{GE}^2$, and $\sigma_\varepsilon^2$ represent the genotypic variance, variance of the interaction between the genotype (G) and the environment (E), and the variance of stochastic error, respectively; $n$ represents the number of environments, and $r$ denotes the number of replicates in each environment.

## QTL mapping

One high-density linkage map with 3,294 SLAF markers and 347 SSR markers, and the other genetic map with 166 SSR markers evenly distributed on the genome were constructed for the $BC_1$ and the $F_2$ populations, respectively (S1 and S2 Tables). QTL mapping was performed using the IciMapping 4.1 software [34], with the inclusive composite interval mapping additive (ICIM-ADD) model. The significant LOD threshold was determined based on 1000 permutations with a type 1 error of 0.05. The QTLs with similar genomic locations (within 5 cM) and same direction of additive effects in different environments were regarded as the same QTL and designated the same name [6]. The QTLs that were identified in more than one environment were regarded as stable QTLs, and QTLs with more than 10% of PVE in at least one environment were regarded as major QTLs [6]. A graphical representation of the map was constructed using the Mapchart 2.3 software [35].

## Identification of candidate genes for major QTLs

The marker sequences flanking each QTL were aligned against the reference genome of sesame 'Zhongzhi 13 v2.0' [15] for identifying the positions of the QTLs using TBtools software v1.106 [36]. The genes were extracted from the mapping intervals, and the gene functions were annotated by comparing the gene sequence with non-redundant protein sequence database.

## Results

### Phenotype description of seed size and shape traits

The female parent Yuzhi 4 was a large-seed cultivar, while the paternal parent BS was an exotic germplasm line with a significantly smaller seed size (Fig 1A). Compared to that of the Yuzhi 4, which had an average SL, SW, and L/W of 3.24 mm, 1.76 mm, and 1.84, respectively, the seed size of the BS line was smaller, with an average SL, SW, and L/W of 2.32 mm, 1.45 mm, and 1.56, respectively (Fig 1B). The traits related to seed size and shape of the $F_2$ and $BC_1$ populations were measured, and the descriptive statistics are provided in Table 1. The SL, SW, and L/W of the $F_2$ population ranged from 2.20 to 3.00 mm, 1.32 to 1.76 mm, and 1.61 to 1.80, respectively, and the mean values were 2.63 mm, 1.56 mm, and 1.69, respectively, across two different environments. The SL, SW, and L/W of the $BC_1$ population ranged from 2.52 to 3.22 mm, 1.47 to 1.78 mm, and 1.62 to 1.91, respectively, and the mean values were 2.87 mm, 1.62 mm, and 1.78, respectively, across three different environments (Table 1).

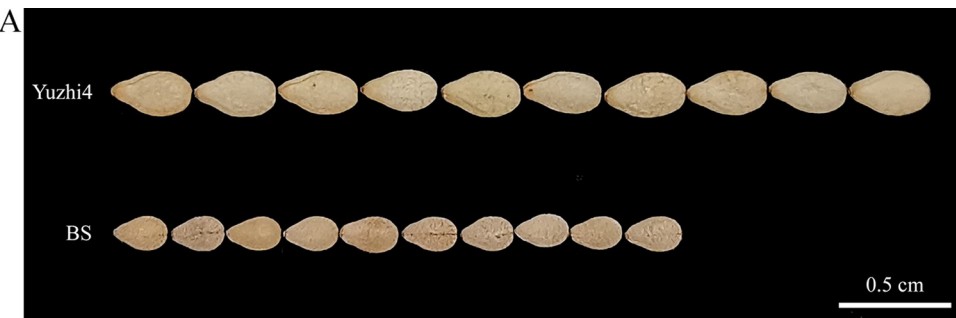

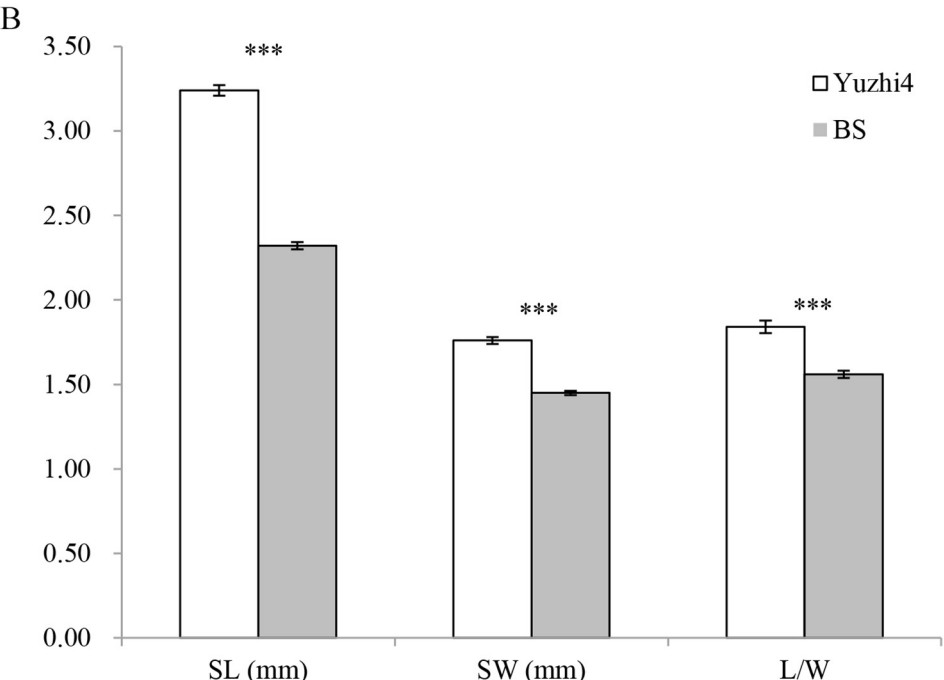

**Fig 1. Phenotypic characterization of the seeds of Yuzhi 4 and BS lines.** (A) Seed size and shape phenotypes of Yuzhi 4 and BS lines; scale bar: 0.5 cm. (B) Comparison of the SL, SW, and L/W ratio of Yuzhi 4 and BS lines. The data are presented as the mean ± standard deviation (SD) (n = 3). The asterisks indicate significant differences (t-test; *P < 0.05; **P < 0.01; ***P < 0.001) between Yuzhi 4 and BS.

**Table 1. Phenotypic variation in the seed size and shape of the two populations across different environments.**

| Population | Trait | Environment | Mean | SD | Range | CV (%) | Excess kurtosis | Skewness |
|---|---|---|---|---|---|---|---|---|
| $F_2$ | SL (mm) | 2019SY | 2.56 | 0.13 | 2.20–2.93 | 5.08 | -0.38 | -0.20 |
| | | 2020SY | 2.70 | 0.12 | 2.38–3.00 | 4.44 | -0.16 | -0.30 |
| | | Mean | 2.63 | 0.1 | 2.34–2.88 | 3.8 | -0.39 | -0.14 |
| | SW (mm) | 2019SY | 1.52 | 0.07 | 1.32–1.69 | 4.61 | -0.42 | -0.11 |
| | | 2020SY | 1.61 | 0.07 | 1.45–1.76 | 4.35 | -0.36 | -0.15 |
| | | Mean | 1.56 | 0.05 | 1.43–1.69 | 3.21 | -0.54 | 0.01 |
| | L/W | 2019SY | 1.70 | 0.04 | 1.61–1.80 | 2.35 | -0.25 | 0.20 |
| | | 2020SY | 1.69 | 0.04 | 1.59–1.78 | 2.37 | -0.57 | -0.08 |
| | | Mean | 1.69 | 0.03 | 1.61–1.78 | 1.78 | -0.29 | 0.09 |
| $BC_1$ | SL (mm) | 2018LH | 2.92 | 0.11 | 2.60–3.15 | 3.77 | -0.11 | -0.37 |
| | | 2018NY | 2.90 | 0.12 | 2.56–3.22 | 4.14 | -0.11 | -0.04 |
| | | 2018PY | 2.79 | 0.10 | 2.52–3.06 | 3.58 | 0.16 | -0.08 |
| | | Mean | 2.87 | 0.09 | 2.66–3.14 | 3.14 | -0.37 | 0.18 |
| | SW (mm) | 2018LH | 1.65 | 0.06 | 1.49–1.77 | 3.64 | -0.30 | -0.29 |
| | | 2018NY | 1.63 | 0.06 | 1.47–1.78 | 3.68 | 0.08 | -0.08 |
| | | 2018PY | 1.57 | 0.04 | 1.48–1.67 | 2.55 | -0.71 | 0.15 |
| | | Mean | 1.62 | 0.04 | 1.52–1.73 | 2.47 | -0.19 | 0.08 |
| | L/W | 2018LH | 1.77 | 0.04 | 1.65–1.90 | 2.26 | 0.89 | 0.01 |
| | | 2018NY | 1.78 | 0.05 | 1.62–1.91 | 2.81 | 0.89 | -0.15 |
| | | 2018PY | 1.78 | 0.04 | 1.68–1.90 | 2.25 | 0.33 | 0.18 |
| | | Mean | 1.78 | 0.04 | 1.67–1.88 | 2.25 | 0.61 | 0.11 |

Note: SD, standard deviation; CV, coefficient of variation.

Analysis of the skewness and kurtosis indicated that the three traits followed a normal distribution pattern in the two populations. The results of the analysis of variance indicated that the effects of genotypes (G) and environments (E) were highly significant in the $BC_1$ population ($P < 0.01$); however, the interaction between G and E (G × E) did not significantly affect the SL, SW, and L/W across the three environments (S3 Table). The broad-sense heritability of the three traits was relatively high, ranging from 0.73 to 0.88 (S3 Table). Furthermore, for each of the traits, a significantly positive correlation of the phenotypic values between each of environments in the $BC_1$ population. These findings indicated that genetics plays a major role in regulating the size and shape of sesame seeds. The results of phenotypic correlation indicated that the SL was significantly positively correlated with the SW ($r = 0.87$, 2019SY; $r = 0.86$, 2020SY; $P < 0.001$) and L/W ($r = 0.42$, 2019SY; $r = 0.47$, 2020SY; $P < 0.001$) in the $F_2$ population; however, no significant correlation was found between the SW and L/W. The SL was significantly positively correlated with the SW and L/W in the $BC_1$ population, and the correlation coefficients were 0.76–0.81 and 0.45–0.60 ($P < 0.001$), respectively, across the three environments. The SW was significantly negatively correlated with the L/W in 2018LH environment ($r = -0.16$, $P < 0.05$), while no significant correlation was found in the other two environments (S4 Table).

## QTLs detected in the $BC_1$ population

A high-density linkage map with 3,294 SLAF markers and 347 SSR markers was constructed for the $BC_1$ population (S1 Table). This map contained 13 linkage groups (LGs), and covered a total of 1266.87 cM genetic distance. A total of 39 QTLs, distributed on 11 LGs, were detected for SL, SW, and L/W in the $BC_1$ population across the three environments (Fig 2 and Table 2).

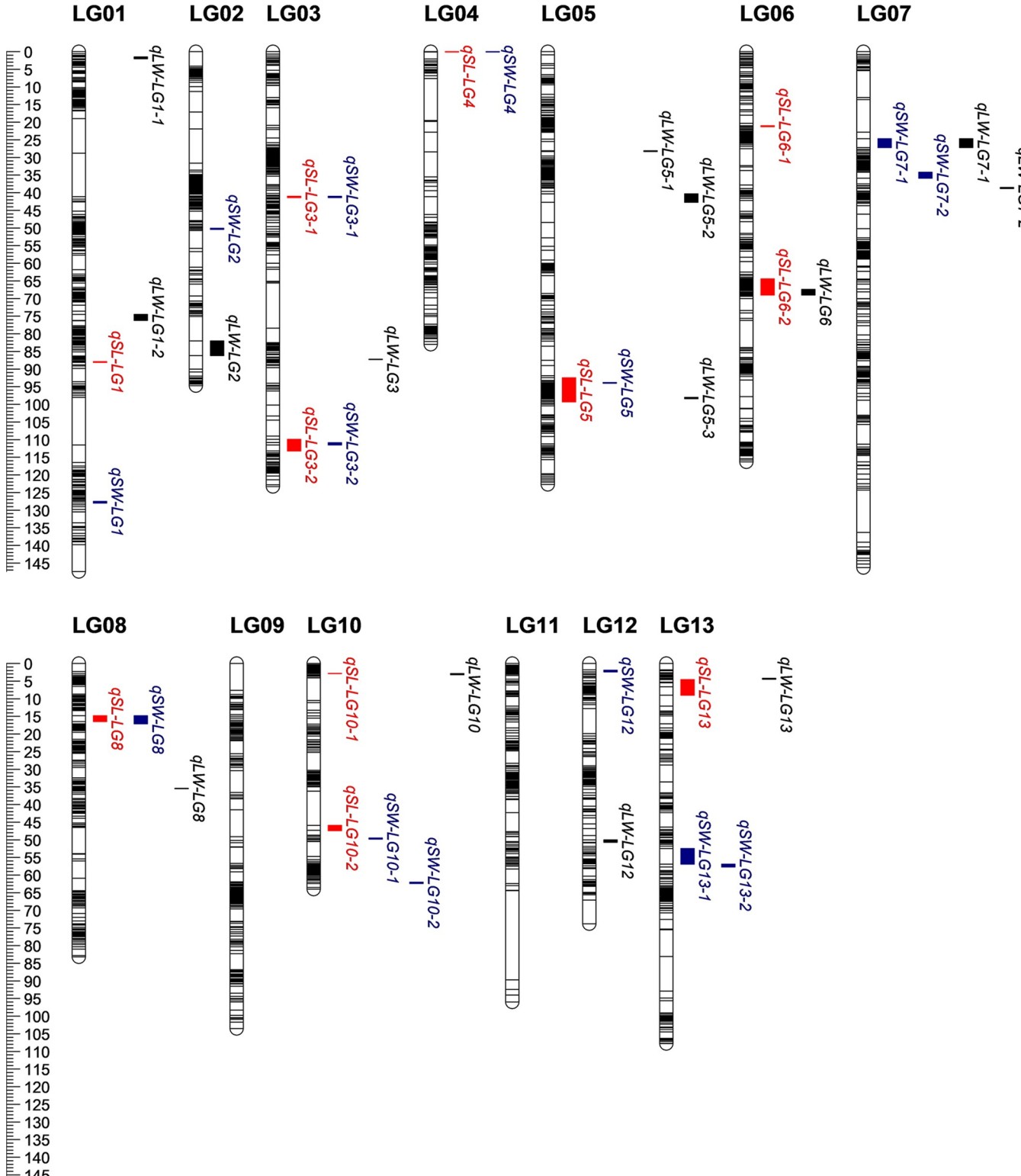

**Fig 2. QTLs for traits related to seed size and shape detected in BC₁ population.** The black bar on each LG column indicates a marker. The QTLs for SL, SW, and L/W are shown in red, blue, and black, respectively.

**Table 2. QTLs for seed size and shape traits identified in the BC$_1$ population across three different environments.**

| Trait | QTL | LG | Position | Flanking Markers | LOD score | PVE (%) | Additive effect | Environment |
|---|---|---|---|---|---|---|---|---|
| SL | qSL-LG1 | LG1 | 87.90 | Marker3181998–Marker3181685 | 3.53 | 3.68 | -0.05 | 2018NY |
| | qSL-LG3-1 | LG3 | 41.30 | Marker83756–Marker113316 | 3.51 | 5.06 | 0.04 | 2018PY |
| | qSL-LG3-2 | LG3 | 109.90 | HSRC0378–HSRC0382 | 5.60 | 5.93 | 0.06 | 2018NY |
| | | LG3 | 109.90 | HSRC0378–HSRC0382 | 3.49 | 4.94 | 0.04 | 2018PY |
| | | LG3 | 111.60 | Marker93889–Marker55215 | 5.65 | 13.52 | 0.07 | 2018LH |
| | qSL-LG4 | LG4 | 0.00 | Marker3297566–Marker3261588 | 7.76 | 8.51 | -0.07 | 2018NY |
| | qSL-LG5 | LG5 | 92.50 | Marker915637–Marker785275 | 7.68 | 8.52 | 0.07 | 2018NY |
| | | LG5 | 94.00 | Marker836389–Marker797497 | 3.81 | 8.87 | 0.06 | 2018LH |
| | | LG5 | 99.00 | Marker953444–Marker763750 | 6.87 | 10.52 | 0.06 | 2018PY |
| | qSL-LG6-1 | LG6 | 21.30 | Marker348764–HSRC4555 | 2.89 | 6.59 | 0.05 | 2018LH |
| | qSL-LG6-2 | LG6 | 64.60 | Marker1958174–Marker1916997 | 5.50 | 8.14 | 0.05 | 2018PY |
| | | LG6 | 69.00 | Marker2066119–HSRC3298 | 4.13 | 9.62 | 0.06 | 2018LH |
| | | LG6 | 69.00 | Marker2066119–HSRC3298 | 10.57 | 12.17 | 0.08 | 2018NY |
| | qSL-LG8 | LG8 | 16.20 | Marker1759630–Marker1772459 | 10.27 | 12.06 | 0.08 | 2018NY |
| | qSL-LG10-1 | LG10 | 2.90 | Marker1051746–Marker1022441 | 7.05 | 10.66 | 0.06 | 2018PY |
| | qSL-LG10-2 | LG10 | 45.90 | Marker1217531–Marker1224607 | 3.45 | 7.91 | 0.06 | 2018LH |
| | qSL-LG13 | LG13 | 4.70 | HSRC0772–Marker1452916 | 5.64 | 5.98 | 0.06 | 2018NY |
| | | LG13 | 8.40 | Marker1553119–Marker1417949 | 5.35 | 7.94 | 0.05 | 2018PY |
| SW | qSW-LG1 | LG1 | 127.90 | Marker540426–Marker553109 | 2.99 | 6.30 | -0.03 | 2018LH |
| | qSW-LG2 | LG2 | 50.40 | Marker2338098–Marker2249851 | 4.11 | 2.08 | -0.02 | 2018PY |
| | qSW-LG3-1 | LG3 | 41.00 | Marker83756–Marker113316 | 10.93 | 6.07 | 0.03 | 2018PY |
| | qSW-LG3-2 | LG3 | 111.50 | HSRC0382–Marker93889 | 5.03 | 10.93 | 0.04 | 2018LH |
| | qSW-LG4 | LG4 | 0.00 | Marker3297566–Marker3261588 | 6.10 | 11.45 | -0.04 | 2018NY |
| | qSW-LG5 | LG5 | 94.00 | Marker836389–Marker797497 | 3.35 | 7.18 | 0.03 | 2018LH |
| | qSW-LG7-1 | LG7 | 24.70 | Marker2501417–HSRC2127 | 2.92 | 1.44 | 0.02 | 2018PY |
| | qSW-LG7-2 | LG7 | 34.20 | Marker2617027–Marker2485692 | 3.48 | 6.26 | 0.03 | 2018NY |
| | qSW-LG8 | LG8 | 16.50 | Marker1759630–Marker1772459 | 7.52 | 14.47 | 0.05 | 2018NY |
| | | LG8 | 16.60 | Marker1772459–HSRC3916 | 11.02 | 6.17 | 0.03 | 2018PY |
| | qSW-LG10-1 | LG10 | 49.70 | Marker1219973–Marker1225793 | 3.82 | 8.16 | 0.03 | 2018LH |
| | qSW-LG10-2 | LG10 | 62.40 | Marker1272663–Marker3048400 | 5.96 | 3.12 | 0.02 | 2018PY |
| | qSW-LG12 | LG12 | 2.40 | Marker665702–Marker3196754 | 3.34 | 1.65 | 0.02 | 2018PY |
| | qSW-LG13-1 | LG13 | 52.60 | Marker1527317–HSRC0602 | 31.77 | 25.29 | 0.07 | 2018PY |
| | qSW-LG13-2 | LG13 | 57.60 | HSRC0602–Marker1498679 | 22.83 | 15.64 | -0.06 | 2018PY |
| L/W | qLW-LG1-1 | LG1 | 1.60 | Marker432643–HSRC2751 | 3.00 | 3.32 | 0.02 | 2018NY |
| | qLW-LG1-2 | LG1 | 74.50 | Marker621883–Marker489568 | 3.85 | 4.41 | 0.02 | 2018PY |
| | qLW-LG2 | LG2 | 86.10 | Marker2376168–Marker2235621 | 5.51 | 6.59 | 0.02 | 2018PY |
| | qLW-LG3 | LG3 | 87.30 | Marker43296–Marker210709 | 3.26 | 3.81 | 0.02 | 2018PY |
| | qLW-LG5-1 | LG5 | 28.10 | Marker848476–Marker758702 | 5.30 | 2.46 | -0.03 | 2018LH |
| | qLW-LG5-2 | LG5 | 40.30 | Marker798154–Marker730491 | 3.50 | 3.80 | -0.02 | 2018NY |
| | qLW-LG5-3 | LG5 | 98.00 | Marker940179–HSRC1253 | 4.87 | 5.94 | 0.02 | 2018PY |
| | | LG5 | 98.30 | Marker757282–Marker685311 | 4.10 | 4.47 | 0.02 | 2018NY |
| | qLW-LG6 | LG6 | 67.40 | Marker2008624–HSRC3391 | 18.25 | 26.79 | 0.04 | 2018PY |
| | | LG6 | 69.00 | Marker2066119–HSRC3298 | 33.74 | 25.37 | 0.08 | 2018LH |
| | | LG6 | 69.00 | Marker2066119–HSRC3298 | 25.35 | 39.36 | 0.06 | 2018NY |
| | qLW-LG7-1 | LG7 | 24.70 | Marker2501417–HSRC2127 | 3.04 | 1.36 | -0.02 | 2018LH |
| | qLW-LG7-2 | LG7 | 38.80 | HSRC2216–Marker2675760 | 4.96 | 5.45 | -0.02 | 2018NY |
| | qLW-LG8 | LG8 | 35.40 | Marker1909531–Marker1829971 | 2.85 | 3.05 | 0.02 | 2018NY |

*(Continued)*

**Table 2.** (Continued)

| Trait | QTL | LG | Position | Flanking Markers | LOD score | PVE (%) | Additive effect | Environment |
|-------|-----|-----|----------|------------------|-----------|---------|-----------------|-------------|
| | *qLW-LG10* | LG10 | 3.10 | Marker1022441–Marker1042635 | 3.50 | 4.06 | 0.02 | 2018PY |
| | *qLW-LG12* | LG12 | 50.50 | Marker1668987–Marker1712748 | 5.82 | 6.54 | 0.02 | 2018NY |
| | *qLW-LG13* | LG13 | 4.20 | Marker1554691–Marker1366199 | 4.78 | 5.56 | 0.02 | 2018PY |

A total of 11 QTLs were identified for SL on the LG1, LG3, LG4, LG5, LG6, LG8, LG10, and LG13, which individually explained 3.68–13.52% of the phenotypic variations, and had LOD scores ranging from 2.89 to 10.27. Five QTLs were identified as major QTLs (*qSL-LG3-2*, *qSL-LG5*, *qSL-LG6-2*, *qSL-LG8*, and *qSL-LG10-1*). The major QTLs, *qSL-LG3-2*, *qSL-LG5*, and *qSL-LG6-2*, which were detected in three environments, were considered as stable major QTLs, and accounted for 4.94–13.52%, 8.51–10.52%, and 8.14–12.17% of the phenotypic variations, respectively.

The PVE of the QTLs ranged from 1.36% to 39.36%, with LOD scores ranging from 2.85 to 33.74. Of the 39 QTLs, 11 were identified as major QTLs which explained more than 10% of the phenotypic variation in at least one environment; and 4, 3, and 32 QTLs were identified in three, two, and one environment, respectively.

A total of 14 QTLs for SW were mapped on the LG1, LG2, LG3, LG4, LG5, LG7, LG8, LG10, LG12, and LG13, which individually explained 1.44–25.29% of the phenotypic variations, and had LOD scores ranging from 2.92% to 31.77%. Of these 14 QTLs, one major QTL, *qSW-LG8*, which had a PVE of 6.17–14.47%, was detected in two environments, and four major QTLs, including *qSW-LG3-2*, *qSW-LG4*, *qSW-LG13-1*, and *qSW-LG13-2*, were identified in one environment, and explained 10.93%, 11.45%, 25.29%, and 15.64% of the phenotypic variations, respectively.

A total of 14 QTLs were detected for L/W, which were distributed on the LG1, LG2, LG3, LG5, LG6, LG7, LG8, LG10, LG12, and LG13, and accounted for 1.36–39.36% of the phenotypic variations, with LOD scores of 2.85–33.75. Of these QTLs, only one major QTL, *qLW-LG6*, was identified in three environments, which explained 18.25–33.74% of the phenotypic variations.

## QTLs identified in the $F_2$ population

A genetic map was constructed using 166 SSR markers for the $F_2$ population. Thirteen QTLs were identified in the 2019SY and 2020SY environments, including 5, 4, and 4 QTLs for SL, SW, and L/W, respectively (S1 Fig and S5 Table). The PVE of the QTLs for SL, SW, and L/W ranged from 6.26% to 20.94%, 4.72% to 21.37%, and 6.71% to 21.64%, respectively. Of these QTLs, the PVE of eight QTLs was higher than 10% in at least one environment, and six QTLs were detected in both environments. The QTLs *qSL-LG3-$F_2$* and *qSL-LG13-$F_2$* were identified for SL, which explained 11.05–13.27% and 6.84–10.42% of the phenotypic variations, respectively, in the two environments. The QTL *qSL-LG10-$F_2$* for SL explained 20.94% of the phenotypic variation in 2020SY environment. The QTL *qSW-LG3-$F_2$* was identified for SW, which had a PVE of 14.88–16.35% in the two environments. The QTL *qSW-LG10-$F_2$* for SW explained 21.37% of the phenotypic variation in 2020SY environment. The QTL *qLW-LG6-$F_2$* for the L/W in the two environments explained 4.96–21.64% of the phenotypic variations. The QTL *qLW-LG7-$F_2$* for L/W had a PVE of 12.74% in 2019SY environment, and the QTL *qLW-LG11-$F_2$* for L/W had a PVE of 11.54% in 2020SY environment.

## Pleiotropic QTLs and co-localized loci

The QTLs for different traits that were located in close vicinity (< 5 cM) or in identical regions were regarded as pleiotropic QTLs. After integrating the two genetic maps, a total of 13 co-

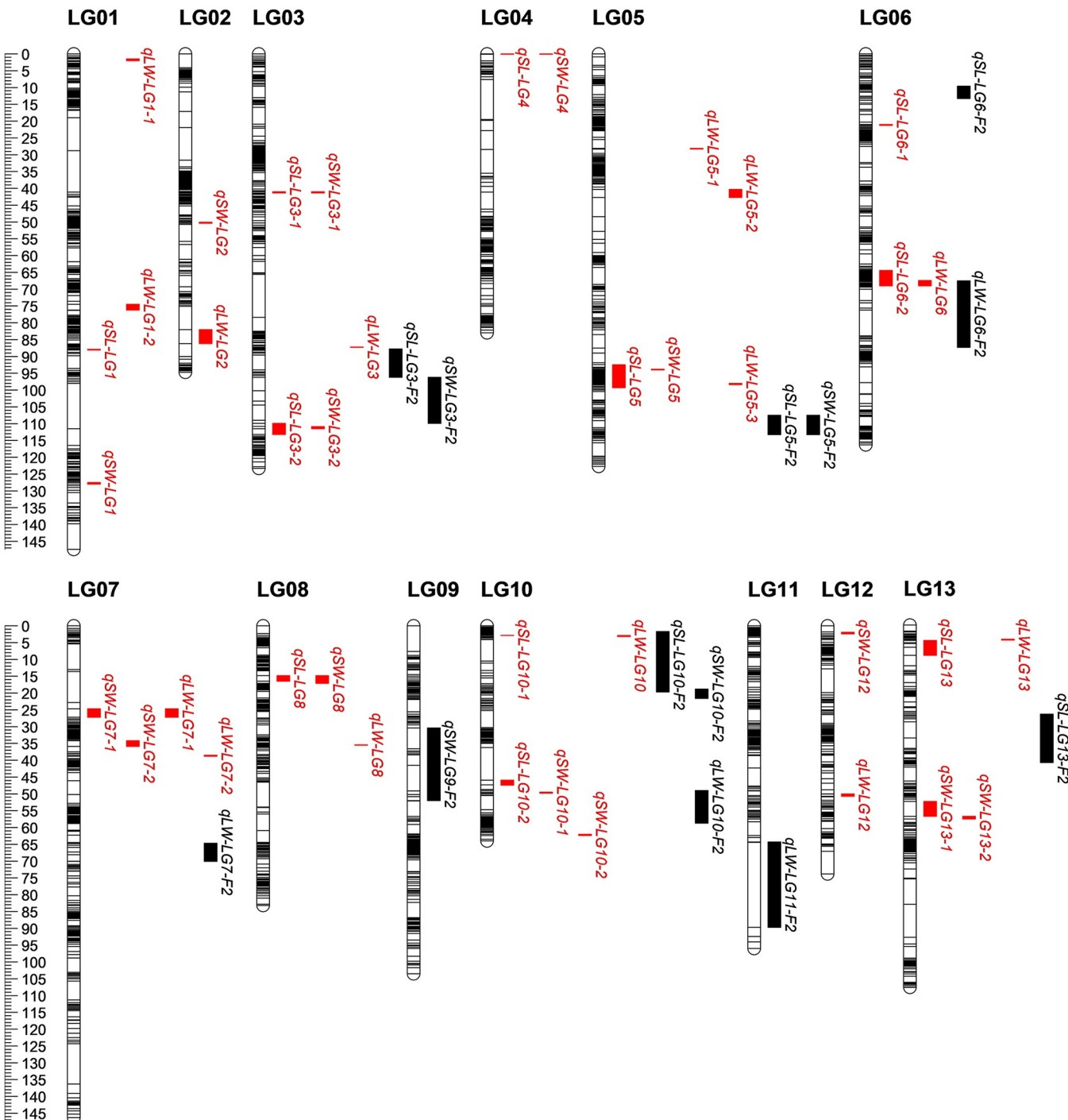

**Fig 3. QTLs for traits related to seed size and shape detected in F$_2$ and BC$_1$ populations.** The black bar on each LG column indicates a marker. The QTLs for F$_2$ and BC$_1$ populations are shown in black and red, respectively.

localized loci were identified to harbor pleiotropic QTLs (Fig 3). Notably, three co-localized loci (locus1-3, locus3-1, and locus9-1) on LG3, LG5, and LG6 harboring stable major QTLs were identified (S6 Table). The locus1-3 was mapped at 96.20–113.24 cM on LG3 in the linkage map and harbored the major QTLs *qSL-LG3-2*, *qSW-LG3-F$_2$*, and *qSW-LG3-2* that were

identified in three, two, and one environment, respectively, explaining 4.94–16.35% of the phenotypic variations. The locus1-3 covered a 2.08 Mb region (16.59–18.67 Mb) on chr1 by the flanking markers HRSC0357 and Marker55215. The locus3-1 was located at 92.45–99.31 cM on LG5 in the linkage map and spanned 1.82 Mb (19.44–21.26 Mb) on chr3 by the flanking markers Marker785275 and Marker763750. The locus3-1 harbored one major QTL *qSL-LG5* and two minor QTLs *qLW-LG5-3* and *qSW-LG5*. The major *qSL-LG5* with PVE of 8.52–10.52% was consistently identified in three environments, while the minor QTLs *qLW-LG5-3* and *qSW-LG5* were detected in two and one environments, respectively. The locus9-1 was located in 64.43–87.31 cM on LG6 in the linkage map and spanned 3.27 Mb with flanking markers HSRC3275 and Marker1958174 on chr9. The locus9-1 harbored the major QTLs *qSL-LG6-2*, *qLW-LG6*, and *qLW-LG6-$F_2$* that were identified in three, three, and two environments, respectively, and explained 8.14–39.36% of the phenotypic variations.

### Functional annotation of the three co-localized QTL regions

In order to identify the genes that potentially regulated seed size and shape, a total of 872 genes located in the three important co-localized loci were identified, including genes that encoded transcription factors, enzymes, and transporters (S7 Table). Gene annotation and orthologous gene analysis indicated that six genes, SIN_1008192, SIN_1015854, SIN_1015831, SIN_1024196, SIN_1024145, and SIN_1015130, were possibly involved in the development of sesame seeds. Furthermore, sequence analysis detected one non-synonymous SNP in the coding region of SIN_1015854, two non-synonymous SNPs in the coding region of SIN_1015831, and one non-synonymous SNP in the coding region of SIN_1015130 between Yuzhi 4 and BS (S2–S5 Figs). SIN_1015854 was annotated as *AUXIN-REGULATED GENE INVOLVED IN ORGAN SIZE* (*ARGOS*), which is a positive regulator of organ size in plants [37, 38]. *SIN_1015831* is an ortholog of the *HMG1* of *Arabidopsis* and encodes 3-hydroxy-3-methylglutaryl coenzyme A reductase, which is involved in sterol biosynthesis [39]. The SIN_1015130 was annotated to encode the E3 ubiquitin-protein ligase RHF2A, which plays an important role in gametogenesis in *Arabidopsis* [40].

### Discussion

The size and shape of seeds are critical traits of crops, which play important roles in determining the yield of seeds, and adaption to certain environments [7, 41]. The complex genetic basis of seed size and shape, which is regulated by several genes involved in various pathways, has been clearly elucidated in model plants. However, it is largely unknown in sesame. The present study revealed that there are substantial genetic variations in seed size (SL and SW) and shape (L/W) among the $F_2$ and $BC_1$ populations. QTL mapping was conducted to identify 13 QTLs in the $F_2$ population in two environments and 39 QTLs in the $BC_1$ population in three environments. Many more QTLs were identified in the $BC_1$ than in the $F_2$ population, maybe it is because many more markers provide enough information for the $BC_1$ to identify more QTLs than in the $F_2$ population. The significant cross-environment correlations and the high heritability of the three traits within the $BC_1$ population indicated that stable QTLs can be identified in different environments. Generally, stable QTLs are defined as those that are consistently detected across different environments, and are of great value for marker-assisted breeding in varieties adapted to various ecological environments [42]. In this study, five major stable QTLs for SL, two for SW, and two for L/W were consistently identified in at least two environments. Furthermore, three major QTLs in the $F_2$ population were verified by QTLs detected in the $BC_1$ population. *qSW-LG3-$F_2$* in the $F_2$ population was mapped to a region of 16.59–18.35 Mb on chr1 similar to *qSW-LG3-2* (18.42–18.48 Mb on chr1) in the $BC_1$ population. *qLW-LG6-$F_2$*

(4.50–7.65 Mb on chr9) and *qSL-LG10-F₂* (11.63–15.16 Mb on chr13) in the F$_2$ population were overlapped with *qLW-LG6* (6.01–7.65 Mb on chr9) and *qSL-LG10-1* (14.81–14.82 Mb on chr13) in BC$_1$ population, respectively. These stable major QTLs (*qSW-LG3-F₂/qSW-LG3-2*, *qLW-LG6-F₂/qLW-LG6*, and *qSL-LG10-F₂/qSL-LG10-1*) in two different populations implied the reliability of the QTLs in this study and the importance of these regions in the genetic improvement of seed size and shape in sesame.

Three co-localized loci (locus1-3, locus3-1, and locus9-1) were identified as important owing to the presence of stable major QTLs for seed size and shape with high PVEs and good stabilities. The QTL *qSL-LG3-2* was located in the interval of 18.35–18.67 Mb on chr1, which co-located with *qSW-LG3-2* (18.42–18.48 Mb) and *qSW-LG3-F₂* (16.59–18.35 Mb). This co-localized locus, spanning 2.08 Mb (16.59–18.67 Mb) on chr1, offered a high level of contribution to PVE by these three QTLs, i.e., 4.94–13.52% for SL across three environments, 10.93% for SW in one environment, and 14.88–16.35% for SW across two environments. Another co-localized locus was mapped in a 1.4 Mb region (14.6–16.0 Mb) on Chr3, in which *qSL-LG5* (19.44–21.26 Mb) was overlapped with *qSW-LG5* (19.49–20.65 Mb) and *qLW-LG5-3* (20.52–21.23 Mb). The QTLs *qSL-LG5* for SL, *qSW-LG5* for SW, and *qLW-LG5-3* for L/W were identified in three, one, and two environments, and explained 8.52–10.52%, 7.18%, and 4.47–10.52% of phenotypic variations, respectively. The third co-localized locus spanned 3.27 Mb (4.50–7.77 Mb) on chr9, in which *qSL-LG6-2* (6.00–7.77 Mb) co-located with *qLW-LG6* (6.01–7.65 Mb) and *qLW-LG6-F₂* (4.50–7.65 Mb). This co-localized locus offered a high level of contribution to PVE by these three QTLs, i.e., 8.14–12.17% for SL across three environments, 25.37–39.36% for L/W across three environments, and 9.88–21.64% for L/W across two environments. In conclusion, these three important co-localized loci were associated with stable major QTLs for SL, SW, and L/W. The application of genetic markers in these loci to breeding programs can potentially optimize the selection of multiple traits related to sesame seed size and shape.

Several QTLs related to seed weight have been identified in previous studies by linkage mapping or association analysis [6, 19, 21, 43]. The seed weight is considerably affected by the seed size and shape. In order to determine the genetic relationships between seed weight and seed size/shape at the individual QTL level, we compared the physical genomic locations of the QTLs identified in this study with QTLs for seed weight in previous studies. The interval regions of locus1-3 (harboring *qSL-LG3-2*, *qSW-LG3-2*, and *qSW-LG3-F₂*), locus3-1 (harboring *qSL-LG5*, *qSW-LG5*, and *qLW-LG5-3*), locus4-1 (harboring *qSL-LG8-2* and *qSW-LG8-2*), and locus13-1 (harboring *qSW-LG10-3* and *qLW-LG10-F₂*) overlapped with those of *qSW_LG03*, *qSW_LG05-2*, *qSW_LG08-1*, and *qSW_LG10*, respectively, reported in the study by Mei et al. [6]. The QTLs *qSL-LG6-1* and *qSL-LG6-F₂* were located in 17.54–17.67 Mb and 19.63–22.67 Mb, respectively, and overlapped with the QTL *Qtgw-11* detected by Wu et al. [19]. The results indicated that the seed size/shape strongly influenced the seed weight at the QTL level, and these QTLs should be selected for marker-assisted selection in breeding programs for improving the yield of sesame by increasing the seed weight. Based on the above results, candidate genes and causative sites for these important traits will be identified by QTL fine-mapping or GWAS. With more genes that underlie quantitative traits identified, navigation breeding will be applied in sesame, which has been successfully used in rice [44].

A total of 872 genes, located in the three important co-localized QTL regions, were identified in this study. Gene annotation, orthologous gene analysis, and sequence analysis revealed that *SIN_1015854*, *SIN_1015831*, and *SIN_1015130* are likely related to sesame seed size and shape. The gene *SIN_1015854* was annotated as *ARGOS*, which is highly induced by auxin and partakes in regulating organ size in *Arabidopsis* [37]. The overexpression of *ARGOS* or *BrARGOS* in *Arabidopsis* leads to the development of larger organs owing to enhanced cell

proliferation [37, 45]. Wang et al. [38] also reported that the overexpression of the *OsARGOS* gene of rice in *Arabidopsis* increases the size of organs by increasing the number and size of cells. The gene *SIN_1015831* encodes an HMG1 protein, which is an important enzyme in the mevalonate pathway of sterol biosynthesis [39]. Mutations in the *hmg1* gene resulted in dwarfism and short siliques owing to reduced cellular elongation resulting from low sterol levels [39]. *SIN_1015130* is an ortholog of the *RHF2A* of *Arabidopsis* and encodes an E3 ubiquitin-protein ligase. A previous study demonstrated that a *rhf1arhf2a* double mutant developed short siliques and exhibited reduced fertility, resulting from defective gametophyte formation due to mitotic cell cycle arrest [40]. Altogether, the findings of the present study lay a foundation for further fine mapping and map-based cloning of these major QTLs for seed size and shape in sesame.

## Conclusion

To understand the genetic basis of seed size and shape in sesame, we developed a high-density genetic map for the $BC_1$ population with 3,294 SLAF markers and 347 SSR markers, and a genetic map for the $F_2$ population with 166 SSR markers. The sesame seed size (SL and SW) and shape (L/W) of the $F_2$ and $BC_1$ populations were measured, and the broad-sense heritability of the three traits ranged from 0.73 to 0.88. QTL mapping showed that 52 QTLs, including 13 in $F_2$ and 39 in $BC_1$ populations, for the three traits were identified and explained phenotypic variations from 3.68 to 21.64%. After integrating the two genetic maps, thirteen co-localized loci were identified. It is worth noting that three co-localized loci locus1-3, locus3-1, and locus9-1 harboring stable major QTLs were identified. Finally, three candidate genes in the three loci which are likely related to sesame seed size and shape were identified. These results will provide new insights into the genetic basis of seed size and shape, and useful information for breeding strategies to improve the seed yield of sesame.

## Supporting information

**S1 Fig. QTLs for traits related to seed size and shape detected in the $F_2$ population.** The black bar on each LG column indicates a SSR marker. The QTLs for SL, SW, and L/W are shown in red, blue, and black, respectively.
(TIF)

**S2 Fig. Non-synonymous SNPs identified in SIN_1015854, SIN_1015831, and SIN_1015130.**
(TIF)

**S3 Fig. Sequence alignment of SIN_1015854 between Yuzhi 4 and BS.**
(TIF)

**S4 Fig. Sequence alignment of SIN_1015831 between Yuzhi 4 and BS.**
(TIF)

**S5 Fig. Sequence alignment of SIN_1015130 between Yuzhi 4 and BS.**
(PNG)

**S1 Table. Detailed information on the genetic map of the $BC_1$ population.**
(XLSX)

**S2 Table. Detailed information on the genetic map of the $F_2$ population.**
(XLSX)

**S3 Table. Analysis of the broad-sense of heritability for three seed size and shape traits in the BC$_1$ population.**
(XLSX)

**S4 Table. Phenotypic correlation coefficients between the seed size and shape traits in the two populations.**
(XLSX)

**S5 Table. QTLs for seed size and shape traits identified in the F$_2$ population.**
(XLSX)

**S6 Table. The co-localized loci identified in the two populations.**
(XLSX)

**S7 Table. The 872 candidate genes identified in three important co-localized QTL regions in sesame.**
(XLSX)

## Author Contributions

**Conceptualization:** Hongxian Mei, Yongzhan Zheng.

**Data curation:** Chengqi Cui.

**Funding acquisition:** Hongxian Mei, Chengqi Cui, Yanyang Liu, Yongzhan Zheng.

**Investigation:** Yanyang Liu, Zhenwei Du, Ke Wu, Xiaolin Jiang.

**Methodology:** Hongxian Mei.

**Project administration:** Haiyang Zhang.

**Supervision:** Hongxian Mei, Haiyang Zhang.

**Validation:** Haiyang Zhang.

**Writing – original draft:** Hongxian Mei, Chengqi Cui.

**Writing – review & editing:** Hongxian Mei, Yanyang Liu.

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
