## [Decision Letter · Decision Letter 0]

29 Aug 2023

PONE-D-23-23432QTL analysis of traits related to seed size and shape in sesame (Sesamum indicum L.)PLOS ONE

Dear Dr. Zheng,

Thank you for submitting your manuscript to PLOS ONE. After careful consideration, we feel that it has merit but does not fully meet PLOS ONE’s publication criteria as it currently stands. Therefore, we invite you to submit a revised version of the manuscript that addresses the points raised during the review process.

We look forward to receiving your revised manuscript.

Kind regards,

Maoteng Li

Academic Editor

PLOS ONE

Journal Requirements:

Reviewers' comments:

Reviewer's Responses to Questions

**Comments to the Author**

1. Is the manuscript technically sound, and do the data support the conclusions?

Reviewer #1: Yes

Reviewer #2: Yes

2. Has the statistical analysis been performed appropriately and rigorously? 

Reviewer #1: Yes

Reviewer #2: Yes

3. Have the authors made all data underlying the findings in their manuscript fully available?

Reviewer #1: Yes

Reviewer #2: Yes

4. Is the manuscript presented in an intelligible fashion and written in standard English?

Reviewer #1: Yes

Reviewer #2: Yes

5. Review Comments to the Author

Reviewer #1: The Manuscript provides information on QTLs linked to sesame seed size and shape, which is a rare report in the public. However, there were still some minor concerns in the study.

Line 70,72 “SNP” should be “SNPs”.

Line 133, “QTLs with more than 10% of PVE regarded as major QTLs”, What is the basis for this regarded?

Line 260, “sequence analysis detected” is cloning in both parents? If so, the sequencing results are included as Supporting information in the manuscript

The formatting of the references needs to be revised. Add units for traits in Table 1

How identification of the sesame seed size and shape QTLs could be useful in future molecular breeding program?

Based on the gene mapping and QTL analysis, authors found three candidate genes for seed size and shape. Do you have any plan for the complementation study to confirm the gene function？

Reviewer #2: In this manuscript, F2 and BC1 populations were developed by crossing the Yuzhi 4 and Bengal small-seed (BS) lines for detecting the quantitative trait loci (QTLs) of traits related to seed size and shape. A total of 52 QTLs, including 13 in F2 21 and 39 in BC1 populations were identified. Three important co-localized loci were subsequently identified, harboring the stable major QTLs, which may provide useful information for future breeding strategies aimed at improving the seed yield of sesame, some questions and suggestions are as follows:

1 in line 146，’the seed size of the BS line was smaller’, there should be added statistical analysis for the phenotype of the parents.

2 in line 169-171, ‘the phenotypic values of the three traits were significantly positively correlated between environments in the BC1 population.’ The wording here is unclear.

3 the description of linkage map should be added in the results.

4 in line 260-262, non-synonymous SNPs were identified, maybe we can use software to predict whether the SNPs affect the protein function, http://www.ppved.org.cn/index.html. The transcription level of the candidate genes also should be considered.

6. PLOS authors have the option to publish the peer review history of their article (what does this mean?). If published, this will include your full peer review and any attached files.

Reviewer #1: No

Reviewer #2: No

---

## [Author Response · Author response to Decision Letter 0]

24 Sep 2023

Thank you for the reviewers’ comments concerning our manuscript entitled “QTL analysis of traits related to seed size and shape in sesame (Sesamum indicum L.)”. Those comments are all valuable and very helpful for revising and improving our paper, as well as the important guiding significance to our research. We have studied comments carefully and have made correction which we hope meet with approval. Revised manuscript has been uploaded. The main corrections in the paper and the responds to the reviewer’s comments are as flowing:

Reviewer #1: The Manuscript provides information on QTLs linked to sesame seed size and shape, which is a rare report in the public. However, there were still some minor concerns in the study.

1. Line 70,72 “SNP” should be “SNPs”.

We are grateful for the suggestion. Yes，“SNP” has been amended to “SNPs”

2. Line 133, “QTLs with more than 10% of PVE regarded as major QTLs”, What is the basis for this regarded?

Mei et al (2021) reported that QTLs that explained more than 10% of phenotypic variation in at least one environment were considered major QTLs in the materials and methods. In results, they performed QTL mapping for yield‑related traits in sesame, and identified 20 major QTLs that explained more than 10% of the corresponding phenotypic variation in at least one environment. We added this reference in line 133.

Mei H, Liu Y, Cui C, Hu C, Xie F, Zheng L, et al. QTL mapping of yield‑related traits in sesame. Mol Breeding. 2021; 41(7):43. doi:10.1007/s11032-021-01236-x.

3. Line 260, “sequence analysis detected” is cloning in both parents? If so, the sequencing results are included as Supporting information in the manuscript.

Yes, the genes were cloned and sequenced in both parents. The sequencing results were provided as Supporting information (S3-S5 Figs).

4. The formatting of the references needs to be revised. Add units for traits in Table 1. How identification of the sesame seed size and shape QTLs could be useful in future molecular breeding program?

Thank you for the suggestion. We have revised the formatting of the references and added units for traits in Table 1 (line 162). In this study, we identified several sesame seed size and shape QTLs. We will develop molecular markers closely linked to the related traits based on the QTLs, and use these markers to select objective trait. Furthermore, candidate genes and causative sites for these important traits will be identified by QTL fine-mapping or GWAS in sesame. With more genes that underlie quantitative traits identified, navigation breeding will be applied in sesame, which has been successfully used in rice (Wei et al. 2021).

Wei X, Qiu J, Yong K, Fan J, Zhang Q, Hua H, et al. A quantitative genomics map of rice provides genetic insights and guides breeding. Nat Genet. 2021; 53(2):243-253. doi:10.1038/s41588-020-00769-9.

5. Based on the gene mapping and QTL analysis, authors found three candidate genes for seed size and shape. Do you have any plan for the complementation study to confirm the gene function？

Yes, we will analyze the expression patterns of three candidate genes using qRT-PCR, analyze the location of the proteins by subcellular localization, and verify the function of the genes using genetic transformation technology. 

Reviewer #2:

1. in line 146，‘the seed size of the BS line was smaller’, there should be added statistical analysis for the phenotype of the parents.

Thanks for your suggestion. Yes, statistical analysis for the phenotype of the parents is shown in Figure 1B. 

2. in line 169-171, ‘the phenotypic values of the three traits were significantly positively correlated between environments in the BC1 population.’ The wording here is unclear.

We are very sorry for our unclear description. This sentence has been rephrased as follows: for each of the traits, a significantly positive correlation of the phenotypic values between each of environments in the BC1 population. (Line 170-171)

3. the description of linkage map should be added in the results.

Yes, we have added the description of linkage map in manuscript. (Line 185-187; line 220)

4 in line 260-262, non-synonymous SNPs were identified, maybe we can use software to predict whether the SNPs affect the protein function, http://www.ppved.org.cn/index.html. The transcription level of the candidate genes also should be considered.

Yes, we have performed prediction of protein function by the software, and found that SNPs did not affect the protein function. Therefore, we will perform the qRT-PCR to consider the transcription level of the candidate genes, and carry out the subcellular localization and genetic transformation to analyze function of the candidate genes in next steps.

We tried our best to improve the manuscript and made some changes in the manuscript. These changes will not influence the content and framework of the paper. We appreciate for Editors/Reviewers’ warm work earnestly, and hope that the correction will meet with approval.

Once again, thank you very much for your comments and suggestions. 

Sincerely yours

Yongzhan Zheng

---

## [Editor Report · Decision Letter 1]

6 Oct 2023

QTL analysis of traits related to seed size and shape in sesame (Sesamum indicum L.)

PONE-D-23-23432R1

Dear Dr. Zheng,

We’re pleased to inform you that your manuscript has been judged scientifically suitable for publication and will be formally accepted for publication once it meets all outstanding technical requirements.

Kind regards,

Maoteng Li

Academic Editor

PLOS ONE
---

## [Editor Report · Acceptance letter]

24 Oct 2023

PONE-D-23-23432R1 

QTL analysis of traits related to seed size and shape in sesame (*Sesamum* indicum L.) 

Dear Dr. Zheng:

I'm pleased to inform you that your manuscript has been deemed suitable for publication in PLOS ONE. Congratulations! Your manuscript is now with our production department. 

Kind regards, 

on behalf of

Dr. Maoteng Li 

Academic Editor

PLOS ONE